# Enhanced Expression of Human Endogenous Retroviruses, TRIM28 and SETDB1 in Autism Spectrum Disorder

**DOI:** 10.3390/ijms23115964

**Published:** 2022-05-25

**Authors:** Pier-Angelo Tovo, Chiara Davico, Daniele Marcotulli, Benedetto Vitiello, Valentina Daprà, Cristina Calvi, Paola Montanari, Andrea Carpino, Ilaria Galliano, Massimiliano Bergallo

**Affiliations:** 1Department of Public Health and Pediatric Sciences, University of Turin, Piazza Polonia 94, 10126 Turin, Italy; valentina.dapr@yahoo.it (V.D.); cristina.calvi@unito.it (C.C.); paola.montanari@unito.it (P.M.); ilaria.galliano@unito.it (I.G.); 2Division of Child and Adolescent Neuropsychiatry, Department of Public Health and Pediatric Sciences, University of Turin, Piazza Polonia 94, 10126 Turin, Italy; chiara.davico@unito.it (C.D.); daniele.marcotulli@unito.it (D.M.); benedetto.vitiello@unito.it (B.V.); 3Pediatric Laboratory, Department of Public Health and Pediatric Sciences, University of Turin, Piazza Polonia 94, 10126 Turin, Italy; 4Postgraduate School of Pediatrics, University of Turin, Piazza Polonia 94, 10126 Turin, Italy; andrea.carpino@unito.it

**Keywords:** autism spectrum disorder, human endogenous retroviruses, TRIM28, SETDB1, children

## Abstract

Human endogenous retroviruses (HERVs) are relics of ancestral infections and represent 8% of the human genome. They are no longer infectious, but their activation has been associated with several disorders, including neuropsychiatric conditions. Enhanced expression of HERV-K and HERV-H envelope genes has been found in the blood of autism spectrum disorder (ASD) patients, but no information is available on syncytin 1 (SYN1), SYN2, and multiple sclerosis-associated retrovirus (MSRV), which are thought to be implicated in brain development and immune responses. HERV activation is regulated by TRIM28 and SETDB1, which are part of the epigenetic mechanisms that organize the chromatin architecture in response to external stimuli and are involved in neural cell differentiation and brain inflammation. We assessed, through a PCR realtime Taqman amplification assay, the transcription levels of *pol* genes of HERV-H, -K, and -W families, of *env* genes of SYN1, SYN2, and MSRV, as well as of TRIM28 and SETDB1 in the blood of 33 ASD children (28 males, median 3.8 years, 25–75% interquartile range 3.0–6.0 y) and healthy controls (HC). Significantly higher expressions of TRIM28 and SETDB1, as well as of all the HERV genes tested, except for HERV-W-*pol*, were found in ASD, as compared with HC. Positive correlations were observed between the mRNA levels of TRIM28 or SETDB1 and every HERV gene in ASD patients, but not in HC. Overexpression of TRIM28/SETDB1 and several HERVs in children with ASD and the positive correlations between their transcriptional levels suggest that these may be main players in pathogenetic mechanisms leading to ASD.

## 1. Introduction

Autism spectrum disorder (ASD) is a pervasive neurodevelopmental condition characterized by deficits in communication and social interaction, as well as restricted interests and repetitive behaviors, with variable severity and heterogeneous functional impairment [1]. Susceptibility to ASD is mainly explained by genetic factors [2]. A meta-analysis of twin studies concluded that heritability of ASD can be estimated between 64% and 91% [3]. Increasing evidence underlines, however, the importance of environmental factors, particularly during the critical phase of central nervous system development [4,5]. In animal models, the stimulation of the maternal immune response gives rise to neurodevelopmental and behavioral abnormalities in the offspring [6,7], with permanent alterations in the immune system [8,9]. Maternal immune activation leading to pro-inflammatory cytokine production can elicit fetal inflammation with negative effects for the developing brain and possible pathogenetic relevance to ASD [10,11,12,13]. High levels of inflammatory cytokines were actually found in the blood and cerebrospinal fluid of patients with ASD, and abnormal cytokine profiles have been proposed as biomarkers of ASD [14]. Notably, enhanced cytokine expression was noticed not only in peripheral blood mononuclear cells (PBMCs) of children with ASD, but also of their mothers, and this was associated with upregulation of human endogenous retroviruses (HERVs) [15].

HERVs represent 8% of our genome. They are the relics of ancestral retroviral germ cell infections [16]. HERVs maintain the typical retroviral structure with three principal genes: group-associated antigens (*gag*), polymerase (*pol*) and envelope (*env*), flanked between two regulatory long terminal repeats (LTRs). During evolution, the accumulation of mutations blocked the production of infectious virions and most HERVs became inactive. However, some viral sequences are transcribed and a few encode proteins, such as syncytin 1 (SYN1), an envelope protein encoded by HERV-W [17], and syncytin 2 (SYN2), an envelope protein encoded by HERV-FRD [18], which have been co-opted for essential physiological functions. For instance, they are engaged in placental syncytiotrophoblast formation and regulation of crucial immune functions [19,20,21,22]. HERVs are extensively distributed throughout the human genome and can modulate the transcription of close cellular genes [23,24,25]. Their RNAs, through a copy–paste mechanism, can generate novel insertions into the genome and, being sensed as non-self by pattern recognition receptors, (PRRs) elicit inflammatory and immune reactions [23,24,26,27]. Furthermore, some viral proteins, such as the envelope protein encoded by HERV-W and referred to as multiple sclerosis-associated retrovirus (MSRV), can trigger brain inflammation and autoimmunity [28,29,30], while others, such as the syncytins, exhibit intrinsic immunomodulatory properties [20,21,22,31,32]. Several lines of research have evidenced an association between aberrant HERV expressions and inflammatory or autoimmune diseases [33,34,35,36], as well as neurologic and psychologic disorders [37,38,39], supporting an etiopathogenetic role of retroviruses in these conditions.

Activation of HERVs may be regulated by environmental factors via epigenetic mechanisms, such as DNA methylation and heterochromatin-silencing by histone modifications. Krüppel-associated box domain zinc finger proteins (KRAB-ZFPs) are the largest family of transcriptional regulators in the human genome [40]. Tripartite motif containing 28 (TRIM28), also called KAP1 or TIF1-β, is a E3 ligase acting as a nuclear co-repressor of KRAB-ZFPs [41]. SET domain bifurcated histone lysine methyltransferase 1 (SETDB1), also known as ESET, is a methyltransferase with high specificity for the lysine 9 residue of histone H3 [42,43]. Both TRIM28 and SETBD1 represent specific tags for epigenetic transcriptional repression of retroviral sequences [44,45]. Additionally, growing data document their involvement in many aspects of cell homeostasis, in the control of both innate and adaptive immune responses [45,46,47], in neural cell differentiation, and synapse functions [48,49]. Dysregulation of the epigenetic landscape has become an attractive hypothesis to explain neuropsychiatric diseases [50,51], including ASD [52,53], and alterations in DNA methylation patterns have been observed in the brains of ASD individuals [54,55]. Despite the potential involvement of TRIM28 and SETDB1 in triggering and/or maintaining ASD, to the best of our knowledge, no study has explored their expressions in affected patients.

We aimed to assess the transcription levels of *pol* genes of HERV-H, -K, and -W, the three retroviral families most widely studied [16,23,34], of *env* genes of SYN1, SYN2, and MSRV, as well as of TRIM28 and SETDB1 in the whole blood from children with ASD and healthy controls.

## 2. Results

### 2.1. Study Populations

Thirty three children with ASD were enrolled in the study (Table 1). Fifteen of them (45%) also had an intellectual disability, defined by an IQ below 70 with functional impairment. Two children were being treated with valproic acid at a daily dose of 17 mg/Kg and 16 mg/Kg, respectively; one child was on aripiprazole 0.04 mg/Kg/day, and another on cyproheptadine 0.5 mg/Kg/day.

Healthy controls (HC) consisted of two groups, HC1 and HC2: HC1 included 90 healthy children (45 males, median age 4.4 years, IQR 3.3–7.7 years) who had been tested as controls for detection of *pol* genes of HERV-H, HERV-K, and HERV-W in our previous studies. HC2 included 79 healthy children who were investigated for detection of *env* genes of SYN1, SYN2, MSRV and for expression of TRIM28 and SETDB1 (48 males, median age 6.3 years, IQR 4.0–11.3 years).

### 2.2. Influence of Age on Expression of HERVs, TRIM28, and SETDB1

The median age differed in the three groups of children (ASD, HC1, and HC2). In particular, HC2 patients were significantly older than HC1 and ASD patients (*p* = 0.0132 and *p* = 0.0107, respectively), without a statistically significant difference between HC1 and ASD. The transcriptional levels of each gene, however, were not related to age, with no significant correlation between age and expression of each target gene in either the ASD or the HC groups (data not shown).

### 2.3. Expression Levels of the Housekeeping Gene

The transcription levels of the housekeeping gene *GAPDH* were similar in ASD (median, IQR: 21.69, 21.04–22.07) as in HC (HC1: 21.12, 20.85–21.68, and HC2: 21.75, 21.43–22.28).

### 2.4. Expression Levels of the Pol Genes of HERV-H, HERV-K, and HERV-W

The mRNA levels of the *pol* genes of HERV-H and HERV-K differed significantly between the ASD and HC groups, while no significant difference was found for HERV-W (Figure 1). Median values, IQR 25–75%; HERV-H-*pol*: ASD 1.87, 1.43–2.73; HC 1.00, 0.78–1.34 (*p* < 0.001); HERV-K-*pol*: ASD 2.34, 1.75–3.64; HC 1.03, 0.79–1.31 (*p* < 0.0001); HERV-W-*pol*: ASD 1.02, 0.7–1.45; HC 1.00, 0.77–1.29 (*p* = 0.8524).

### 2.5. Expression Levels of Env Genes of SYN1, SYN2, and MSRV

The mRNA levels of *env* genes of SYN1, SYN2, and MSRV were significantly higher in ASD than in HC (Figure 2). Median values, IQR 25–75%: Syncytin 1-*env*: ASD 1.94, 1.41–2.41; HC 0.95–1.17 (*p* < 0.001). Syncytin 2-*env*: ASD 1.76, 1.11–2.15, HC 0.94, 0.78–1.21 (*p*< 0.0001). MSRV-*env*: ASD 1.49, 1.05–1.98; HC: 1.06, 0.81–1.29 (*p* < 0.003).

### 2.6. Expressions of TRIM28 and SETDB1

The transcriptional levels of TRIM28 and SETDB1 were significantly higher in ASD than in HC (Figure 3). Median values, IQR 25–75%: TRIM28: ASD 1.69, 1.16–2.08; HC 0.90, 0.77–1.25. SETDB1: ASD 2.20, 1.61–2.89; HC 0.99, 0.74–1.19.

### 2.7. Correlations between Expressions of HERVs and TRIM28 or SETDB1

In ASD, mRNA levels of all HERV sequences were strongly correlated with the levels of TRIM28 (Figure 4), whereas no significant correlations were found between these variables in HC (Appendix A).

Similar findings were observed for SETDB1, whose mRNA levels significantly correlated with those of every HERV sequence in ASD (Figure 5), while no significant correlations were observed in HC (Appendix A).

## 3. Discussion

These data document an overexpression of several HERV sequences in a sample of children with ASD. The enhanced transcription levels of *pol* genes of HERV-H and HERV-K families are consistent with the upregulation of *env* genes of the same retroviral families observed in PBMCs from autistic children and their mothers by Balestrieri et al. [15]. The normal levels of HERV-W-*pol* mRNAs in our patients may mirror the contrasting results found by the same authors for HERV-W-*env* in their populations [15,56]. In addition, we also noticed, for the first time, higher expressions of env genes of SYN1, SYN2, and MSRV.

The cause(s) of this enhanced transcription of HERVs and its potential clinical significance remains to be elucidated. In vitro and animal studies have shown that TRIM28 e SETDB1 may be potent corepressors of retroviruses [44,45]. Their higher expressions may give rise to enhanced DNA methylation and heterochromatin formation ultimately leading to HERV silencing [44,57]. However, we found higher mRNA levels of TRIM28 and SETDB1 in ASD, suggesting that the enhanced HERV expression cannot be attributable to impaired transcription of TRIM28 or SETDB1 repressors (see below).

Activation of NF-kB and production of pro-inflammatory cytokines can induce HERV transactivation [58] and high concentrations of inflammatory cytokines were observed in the blood and cerebrospinal fluid of autistic patients [14]. As mentioned, abnormal cytokine profiles were found to be associated with upregulation of HERVs in PBMCs from autistic children and their mothers [15]. It must be pointed out that HERVs can, in turn, shape the immune system and induce inflammatory reactions [16,27,34], with activation of the inflammasome [24]. The result may be a vicious circle leading to inflammatory-driven deleterious effects on the developing brain with consequent clinical manifestations of ASD [10,11,12,13].

Exposure of pregnant women to infectious agents induces activation of the maternal immune system with an increased risk for neuropsychiatric disorders in their children, including ASD [12,59,60,61]. The stimulation of the immune system may result in increased production of neuroinflammatory cytokines in the offspring, with consequent insult to the central nervous system, alterations in synaptic protein expression, and abnormal synaptic connectivity, all of which are typically found in ASD [9,62]. Some exogenous viral infections can trigger HERV activation [63,64,65,66,67,68,69], which, for example in the case of CMV, may also be asymptomatic during pregnancy. Recognition of invasive agents by PRRs elicits the production of interferons that, synergistically with inflammatory cytokines, can induce HERV transcription [58].

Regarding single retroviral genes, there is consensus that syncytins play essential roles in placenta formation and embryonic and fetal growth. In particular, SYN1 transcription is selectively preserved in spermatozoa [70] and can bind with its receptor on oocytes [71], presumably to facilitate the fusion between gametes and the first steps of embryonic development, as emerged for HERV-K in preimplantation blastocysts and pluripotent stem cells [72]. SYN1 is expressed in all circulating leucocytes [73]; upon stimulation, it promotes rapid activation of monocytes [22], synthesis of chemokines and cytokines [20,74], and C reactive protein via TLR3/IL-6 pathway [32]. Both SYN1 and MSRV proteins are more expressed in fetal than in healthy adult brain specimens, while in brain diseases their presence is also associated with neuroinflammation [26,75,76,77]. SYN2 shares with SYN1 syncytial and immunological characteristics: both of them target T-cell activation by modulating the stimulatory activity of DCs [31] and SYN2 significantly influences T-cell functions [21]. A large body of literature shows that MSRV-*env* can trigger brain inflammation and autoimmunity [28,29,76,78], and clinical trials are in progress using an anti-MSRV-*env* monoclonal antibody in patients with autoimmune disorders, such as MS [79] and type 1 diabetes [80]. It is worth mentioning that, on the one side, a familiar history of autoimmunity [81] or a maternal autoimmune disease [82] increases the risk of ASD, while autoantibodies have been detected in affected patients [83,84]. On the other side, the association between HERV overexpression and autoimmune disorders is widely documented [24,26,27,35,36], although whether HERV activation is a cause or an effect of autoimmunity remains questionable. The aberrant expression of HERV elements, in particular of SYN1, SYN2, and MSRV, in ASD children might thus contribute to their immune dysfunctions and to the inflammatory and autoimmune-driven brain damage [10,11,13]. To this purpose, it must be underlined that HEMO, an additional retroviral *env* gene whose protein is synthetized in the placenta and shed into the maternal blood [85], is also upregulated in PBMCs from children with ASD and their mothers [15].

It is worth mentioning that the putative negative effects of HERV activation are expected to occur mainly during the early phases of brain development, while their abnormal mRNA levels were detected at a median age of 3.8 years in our patients. A higher expression of endogenous retroviruses was, however, observed in mouse models of ASD at all ages, from intrauterine life to adulthood [86], supporting a long-lasting activation of endogenous retroviruses that could alter brain function throughout the life span.

The epigenetic control of gene transcription is essential during the embryonic, fetal, and early postnatal growth when cell differentiation and tissue remodeling occurs [87]. Environmental factors induce epigenetic alterations which may ultimately lead to neurodevelopmental disabilities [88]. Multigenerational epigenetic inheritance of DNA methylation marks and chromatic accessibility have been implicated in ASD [89]. A decrease in methylation levels of LINE-1, a transposable element with the ability to self-mobilize throughout the human genome, was found in pediatric patients with ASD [90,91]. TRIM28 is a small ubiquitin-related modifier (SUMO) that, through binding to lysine residues of target proteins, causes their phosphorylation and proteasome-driven degradation. TRIM28 recruits SETDB1 for SUMOylation, a crucial transient post-translational event involved in essential cell functions, such as transcriptional repression, RNA splicing, and protein degradation [92,93]. TRIM28 and SETDB1 exert relevant regulatory activities on both innate and adaptive immune responses [94,95,96]. Both these molecules control the differentiation of T cells. TRIM28 modulates their expansion into regulatory phenotypes [46,97,98]; it influences the yield of DCs and direct T cell priming toward inflammatory effector cells [99]. Furthermore, TRIM28 is highly expressed in the CNS, and its transcriptional control is implicated in human brain evolution and neurological disorders [100]. A gene ontology analysis found that the genes closest to the TRIM28 binding sites are those involved in neuron differentiation [48]. SETDB1 participates in a multitude of biological activities. Among these, it regulates the differentiation of cell lineages within the brain during embryogenesis [51]. It contributes to X chromosome inactivation [101], which may account for the gender bias of some inherited CNS disorders with male predominance, such as ASD [102]. Alterations of SETDB1 or its targeted histone substrate have been associated with the pathogenesis of several diseases of the CNS, including ASD [51,103,104,105]. It must, however, be considered that HERV transcription is thought to be downregulated by TRIM28 and SETDB1, whereas we observed positive correlations between their expressions. Actually, TRIM28 and SETDB1 are essential for maintaining endogenous retroviruses in a silent state in murine pluripotent stem cells and early embryos [57,106]. In contrast, when these cells differentiate in to various somatic cell types, transcription of retroviral sequences is independent of such repressors [57,107], which sometimes may act as transcriptional activators rather than as repressors [97,108]. This might be the case in children with ASD, although other regulatory pathways could account for the parallel changes in the transcription of cellular genes and retroviral sequences. Notably, positive relationships between expressions of TRM28/SETDB1 and HERVs were found in other clinical situations characterized by the activation of the immune system [69]. Its stimulation could trigger the parallel transactivation of both systems, regardless of whether TRIM28/SETDB1 are the main players or not. It must be remembered that the SUMOylation process induced by TRIM28/SETDB and their potential functional interactions with HERVs may be regulated by post-translational events between the encoded proteins, whereas we assessed only their transcriptional profiles.

Valproic acid upregulates transcription of some HERV elements [109] and antipsychotic drugs may trigger epigenetic alterations [110]. No particularly aberrant expressions of HERVs or of TRIM28/SETDB1 were noted in the two patients on valproic acid or in the two on other medications whose impact on these variables is unknown.

The prevalence of ASD has steadily increased for the last decades [111,112]. Prenatal and early postnatal exposure to negative environmental factors, such as pollutants, chemical agents, infections, immune activation, and epigenetics have been associated with ASD [113,114,115]. These environmental factors, e.g., exposure to pesticides [116], maternal cigarette smoking [117], infective agents [63,64,65,66,67,68,69], and immune stimulation [58], exhibit a significant impact on HERV expression. The transcription of both cellular and retroviral genes may be regulated by environmental stimuli of a different nature through epigenetic mechanisms, as those modulated by TRIM28 and SETDB1. Therefore, environmental components responsible for the increasing number of subjects affected by ASD could exert their actions via HERV- and/or TRIM28/SETDB1-driven variations in targeted biologic processes.

In summary, our data provide further support to the potential involvement of HERVs in ASD. The transcriptional alterations of several retroviral sequences suggest that behavioral phenotypes do not derive from the upregulation of a single HERV determinant [15], but from a more extensive and complex process. Provided that the etiopathogenetic role of HERVs in ASD is confirmed by additional studies, the goal to block their activation could be reached with new therapeutic strategies. For instance, combinations of antiretroviral drugs are used for many years in HIV-positive subjects, and their optimal doses and side effects are also well-known in pregnant women and children. HERVs are highly transactivated in HIV+ individuals [118] and antiretroviral therapy reduces the viral burden, not only of HIV, but also of HERVs [119,120,121]. We demonstrated that antiretroviral drugs exhibit a direct anti-proteasome activity [122,123]. The resulting inhibition of NF-kB-driven inflammatory cytokine production can lead to a downregulation of HERV transcription, not only through a targeted action against retroviruses, but also through indirect effects on host cell components. Therefore, the administration of antiretroviral drugs in pregnant women and/or in their children at high risk of ASD may be an exciting hypothesis heralding innovative preventive (and/or therapeutic) interventions.

Environmental stimuli may be translated into the cell as gene expression through the modulation of epigenetic mechanisms. TRIM28/SETDB1, as regulators of the chromatin architecture, are implicated in brain evolution, plasticity, and gene expression. Their high transcription levels in autistic children suggest that they may play an important role in ASD.

Enhanced expressions of HERVs and TRIM28/SETDB1 might represent easy biomarkers for an early diagnosis of ASD. Prospective studies on large series of patients may outline whether they are reliable prognostic markers. The significantly positive correlations between transcripts of TRIM28/SETDB1 and HERVs suggest that the former may exert important regulatory functions on the transactivation of the latter in children with ASD, though the intervention of other underlying regulatory mechanisms acting upstream on both systems cannot be excluded. Given the association between aberrant expressions of epigenetic factors and/or of HERVs in patients with neuropsychiatric disorders, one wonders whether the alterations documented here in ASD are also present, at least in part, in other neurodevelopmental disorders.

## 4. Materials and Methods

### 4.1. Study Populations

ASD patients were recruited at the Division of Child and Adolescent Neuropsychiatry, University of Turin, Regina Margherita Children’s Hospital, Turin, Italy. All patients received a clinical evaluation by a trained Child and Adolescent Neuropsychiatrist and met DSM-5 criteria for ASD [1], further supported by the Autism Diagnostic Observation Schedule [124] or other standardized instruments.

Healthy controls (HC) included asymptomatic children who were tested at the same hospital for routine laboratory examinations and whose results were all within the normal reference range. The study participants did not have evidence of any active medical problems, such as infections, cancer, autoimmune disorders, or neurologic diseases.

### 4.2. Total RNA Extraction

Total RNA was extracted from whole blood using the automated extractor Maxwell (Promega, Madison, WI, USA) following the RNA Blood Kit protocol without modification. This kit provides treatment with DNase during the RNA extraction process. RNA concentration and purity were assessed by traditional UV spectroscopy with absorbance at 260 and 280 nm. The nucleic acid concentration was calculated using the Beer–Lambert law, which predicts a linear change in absorbance with concentration. The RNA concentration range was within manufacturer specifications for the NanoDrop (Thermofisher Scientific, Foster City, CA, USA). UV absorbance measurements were acquired using 1 µL of RNA sample in an ND-1000 spectrophotometer under the RNA-40 settings at room temperature (RT). Using this equation, an A260 reading of 1.0 is equivalent to ~40 µg/mL of single-stranded RNA. The A260/A280 ratio was used to define RNA purity. An A260/A280 ratio of 1.8/2.1 is indicative of highly purified RNA. RNA extracts were directly amplified without reverse transcription to control the genomic DNA contamination. The RNAs were stored at −80 °C until use.

### 4.3. Reverse Transcription

Four hundred nanograms of total RNA was reverse-transcribed with 2 μL of buffer 10× 4.8 μL of MgCl_2_ 25 mM, 2 μL ImpromII (Promega), 1 μL of RNase inhibitor 20 U/L, 0.4 μL random hexamers 250 μM (Promega), 2 μL mix dNTPs 100 mM (Promega), and dd-water in a final volume of 20 μL. The reaction mix was carried out in a GeneAmp PCR system 9700 Thermal Cycle (Applied Biosystems, Foster City, CA, USA) under the following conditions: 5 min at 25 °C, 60 min at 42 °C, and 15 min at 70 °C for the inactivation of the enzyme; the cDNAs were stored at −80° until use.

### 4.4. Transcription Levels of Pol Genes of HERV-H, -K, and -W, of Env Genes of SYN1, SYN2, and MSRV As Well As of TRIM28 and SETDB1 by Real-Time PCR Assay

*GAPDH* was chosen as the reference gene in all determinations, being one of the most stable among reference genes and already used in our previous studies [35,68,69,117]. Relative quantification of mRNA concentrations of *pol* genes of HERV-H, HERV-K, HERV-W, of *env* genes of SYN1, SYN2, and MSRV, as well as of TRIM28, and SETDB1, was achieved by using the ABI PRISM 7500 real-time system (Thermofisher Scientific).

A total of 40 ng of cDNA was amplified in a 20 μL total volume reaction using HERV-H, -K –W mRNA expression kit PP-054, -055, and -056 (BioMole, Turin, Italy) [69]. The PP-BioMole-055 was derived from Schanab et al. [125]. The SYN1-*env* and SYN2-*env* mRNA expressions were also quantified by real-time PCR. A total of 40 ng cDNA was amplified in a 20 μL of total volume reaction containing 2.5 U goTaQ MaterMix (Promega), 1.25 mmol/L MgCl2, 500 nmol of specific primers and 200 nmol of specific probes. The SYN1 primers were: (Sinc1F 5′-ACTTTGTCTCTTCCAGAATCG-3′) (Sinc1R 5′-GCGGTAGATCTTAGTCTTGG-3′), and the probe was: (Sinc1P 6FAM-TGCATCTTGGGCTCCAT-TAMRA) [126]. The Syn 2 primers were: (Sinc2F-GCCTGCAAATAGTCTTCTTT-3′) (Sinc2R- ATAGGGGCTATTCCCATTAG-3′) [127] and the probe was: (Sinc2P-6FAM- TGATATCCGCCAGAAACCTCCC-TAMRA) (this study). The MSRV primers were: (MSRVF 5′-CTTCCAGAATTGAAGCTGTAAAGC-3′) (MSRVR 5′-GGGTTGTGCAGTTGAGATTTCC-3′) and the probe was: (MSRVP 6FAM-TTCTTCAAATGGAGCCCCAGATGCAG-3′-TAMRA) [126]. The probes were designed by Primer ExpressTM software version 3.0 (Applied Biosystems, Foster City, CA, USA).

For TRIM28 and SETDB1, 40 ng of cDNA was amplified using mRNA expression kit PP-044 and PP-045, respectively, (BioMole) in a 20 μL total volume reaction.

The amplifications were run in a 96-well plate at 95 °C for 10 min, followed by 45 cycles at 95 °C for 15 s and at 60 °C for 1 min. Each sample was run in triplicate. Relative quantification of target gene transcripts was performed with the ΔΔCt method. Hence, the fold change was calculated and results were expressed in corresponding arbitrary units, called relative quantification (RQ). Since we measured Ct for every target in all samples, we considered our methods to be suitable for HERV detection and quantification.

### 4.5. Statistical Analysis

The Mann–Whitney test was used to compare the transcripts of pol genes of every HERV family as well as of SYN1, SYN2, TRIM28, and SETDB1 between children with ASD and control children. Spearman’s correlation test was used to evaluate the correlations between transcription levels of each HERV sequence and expressions of TRIM28 or SETDB1 in every group of children. Statistical analyses were done using the Prism software (GraphPad Software, La Jolla, CA, USA). In all analyses, *p* < 0.05 was taken to be statistically significant.

## Figures and Tables

**Figure 1 ijms-23-05964-f001:**
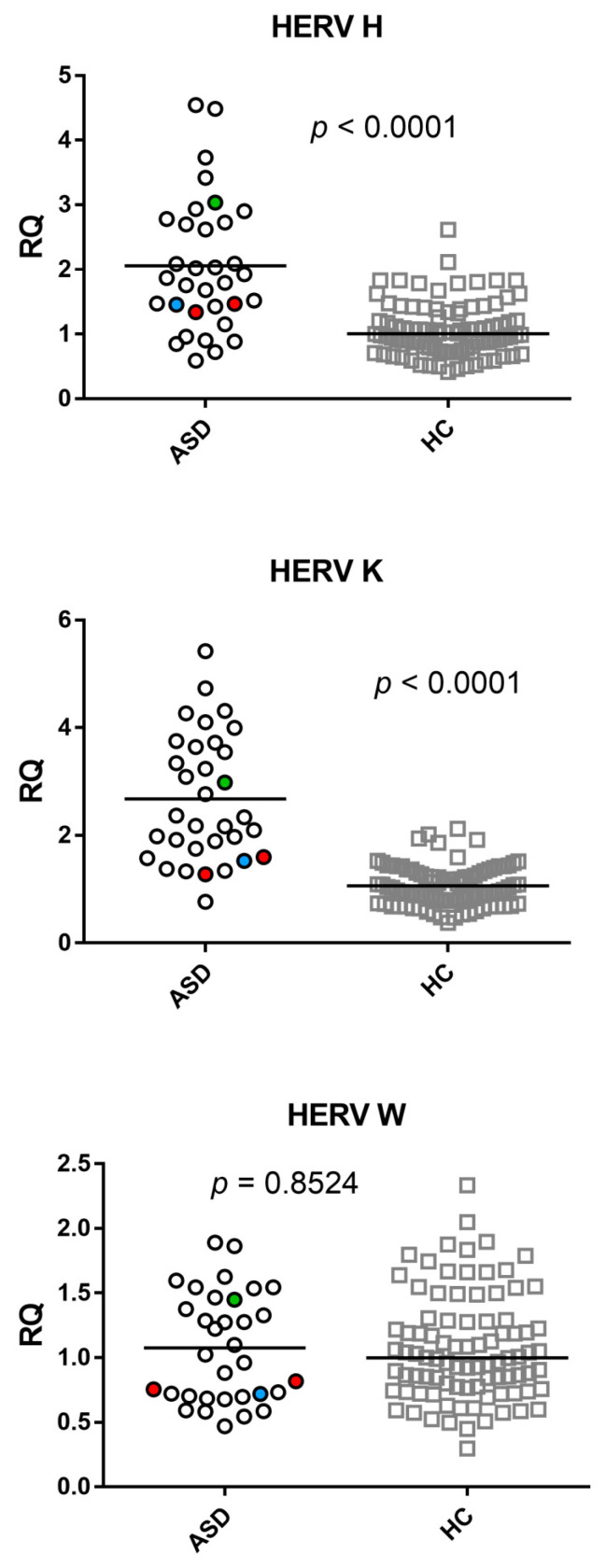
Transcription levels of *pol* genes of HERV-H, HERV-K, and HERV-W, in whole blood from 33 children with autism spectrum disorder (ASD) and 90 healthy controls (HC). RQ: relative quantification. Circles and squares show the median of three individual measurements, horizontal lines the median values. Red circles: values of two patients on valproic acid therapy. Green circle: value of the patient on aripiprazole therapy. Blue circle: value of the patient on cyproheptadine therapy. Statistical analysis: Mann–Whitney test was used to compare the transcriptional levels of each target gene between children with ASD and HC.

**Figure 2 ijms-23-05964-f002:**
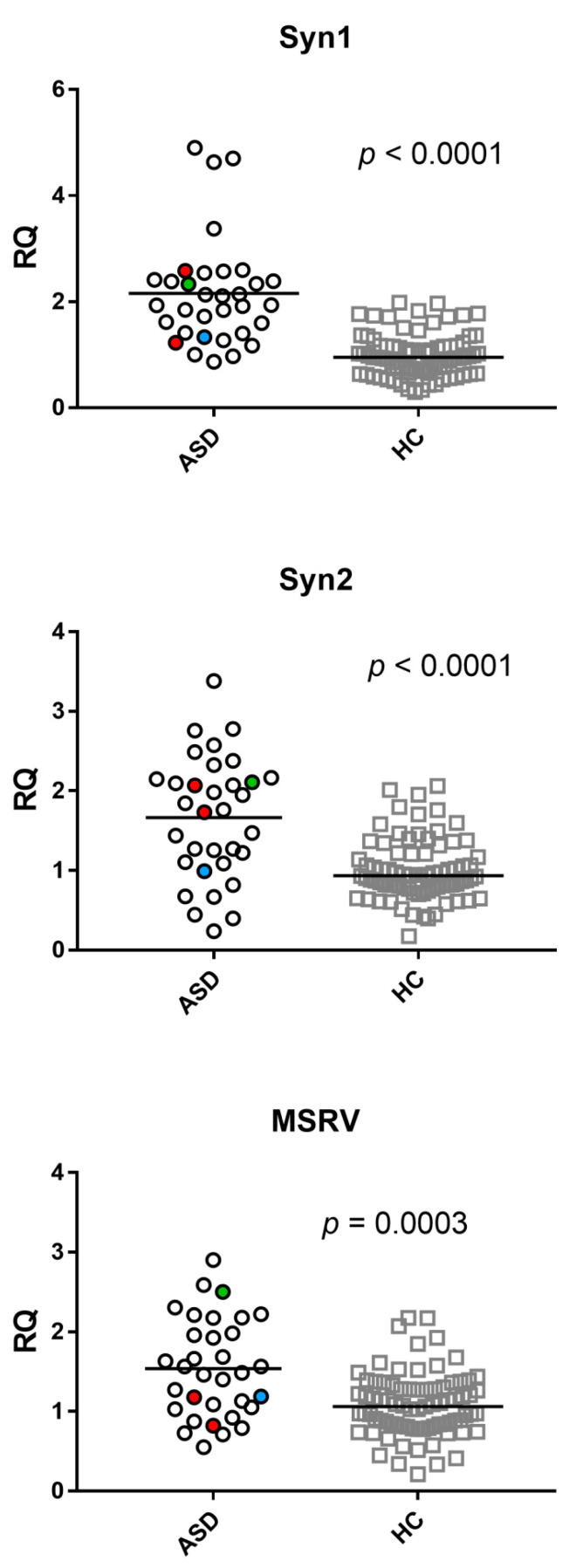
Transcription levels of *env* genes of SYN1, SYN2, and MSRV in whole blood from 33 children with autism spectrum disorder (ASD) and 79 healthy controls (HC). RQ: relative quantification. Circles and squares show the median of three individual measurements, horizontal lines the median values. Red circles: values of two patients on valproic acid therapy. Green circle: value of the patient on aripiprazole therapy. Blue circle: value of the patient on cyproheptadine therapy. Statistical analysis: Mann–Whitney test was used to compare the transcriptional levels of each target gene between children with ASD and HC.

**Figure 3 ijms-23-05964-f003:**
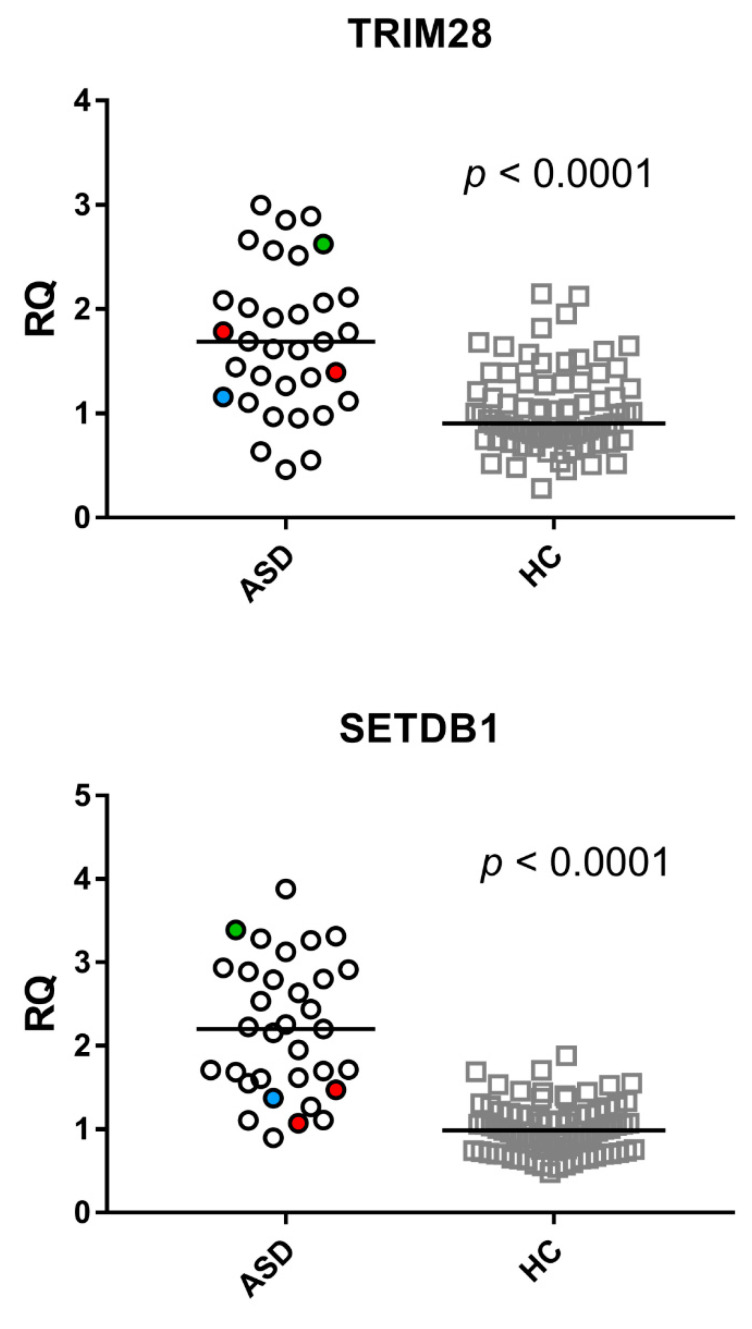
Expression of TRIM28 and SETDB1 in whole blood from 33 children with autism spectrum disorder (ASD) and 79 healthy controls (HC). RQ: relative quantification. Circles and squares show the median of three individual measurements, horizontal lines the median values. Red circles: values of two patients on valproic acid therapy. Green circle: value of the patient on aripiprazole therapy. Blue circle: value of the patient on cyproheptadine therapy. Statistical analysis: Mann–Whitney test was used to compare the transcriptional levels of each target gene between children with ASD and HC.

**Figure 4 ijms-23-05964-f004:**
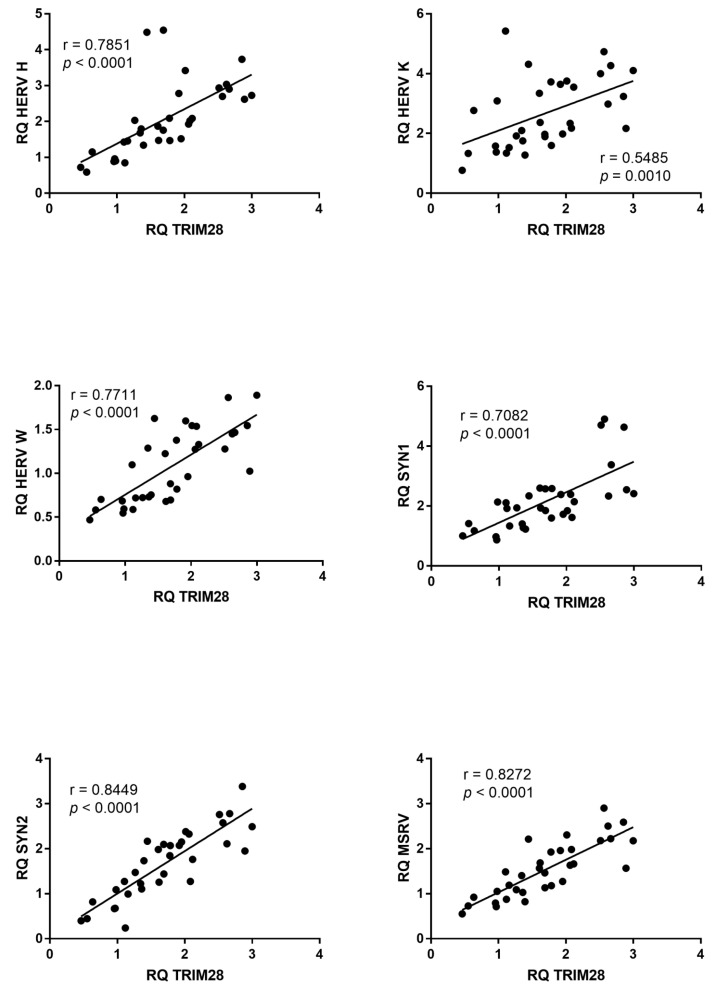
Correlations between transcription levels of TRIM28 and HERV sequences in whole blood from 33 children with ASD. RQ: relative quantification. Circles show the mean of three individual measurements. Line: linear regression line. Statistical analysis: Spearman correlation test.

**Figure 5 ijms-23-05964-f005:**
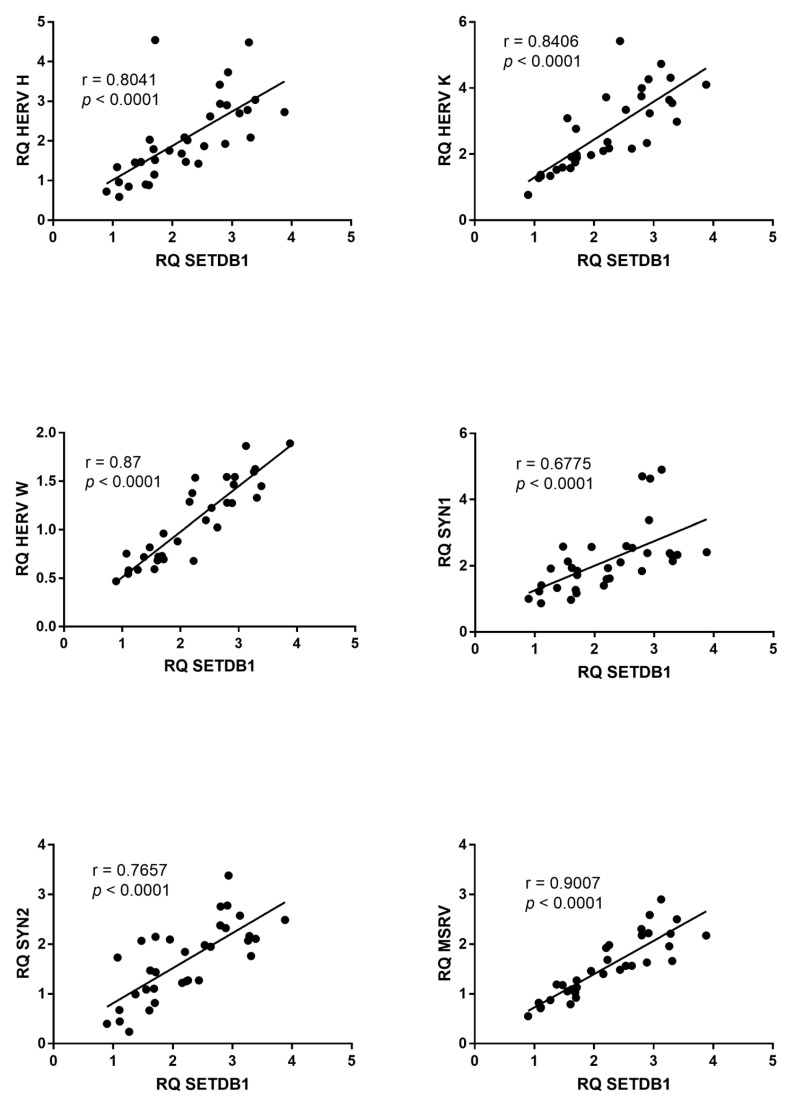
Correlations between transcription levels of SETDB1 and HERV sequences in whole blood from 33 children with ASD. RQ: relative quantification. Circles show the mean of three individual measurements. Line: linear regression line. Statistical analysis: Spearman correlation test.

**Table 1 ijms-23-05964-t001:** Demographics and clinical characteristics of the autism spectrum disorder sample.

Total sample, n	33
Males, n (%)	28 (85)
Age, yr, median (IQR) ^a^	3.8 (3.0–6.0)
Autism severity (ADOS-CSS) ^b^, median (IQR)	7.5 (5.5–8.5) ^c^
Intellectual disability, ^d^ n (%)	15 (45)
Seizures, n (%)	3 (9)
In treatment with valproic acid (%)	2 (6)
In treatment with other psychotropic medication ^e^ (%)	2 (6)

^a^ IQR: Interquartile range 25–75%. ^b^ ADOS-CSS: Autism Diagnostic Observation Schedule—Calibrated Severity criteria Score. ^c^ Based on n = 20. ^d^ Based on DSM5. ^e^ One patient on aripiprazole and another on cyproheptadine.

## Data Availability

Not applicable.

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
