# Peer review of "Enhanced Expression of Human Endogenous Retroviruses, TRIM28 and SETDB1 in Autism Spectrum Disorder"

_ijms, 2022, doi:10.3390/ijms23115964_

Round 1

Reviewer 1 Report

Congratulations to the Authors for their very interesting work. I suggest to better define the clinical characteristics of the group of children with autism (in particular the level of severity of autism based on the tests performed). I also suggest correlating the transcript levels of the studied genes with the cognitive level and severity level of autism (e.g with ADOS score). Last question: Do children on drug therapy (particularly those on VPA therapy) have different transcript levels compared to other children in the autistic group?

Author Response

Reviewer 1.

Congratulations to the Authors for their very interesting work.

We appreciate very much the referee’s interest in our study and her/his positive comment.

  1. I suggest to better define the clinical characteristics of the group of children with autism (in particular the level of severity of autism based on the tests performed).

Following this suggestion, we have summarized the clinical characteristics of the children with autism, specifying presence of intellectual disability, seizure and severity of autism symptoms (see Table 1).

  1. I also suggest correlating the transcript levels of the studied genes with the cognitive level and severity level of autism (e.g with ADOS score).

We totally agree that it could be interesting to evaluate if the transcript levels are linked to the severity of autistic features. To test for the presence of such a correlation, however, would require a larger sample and more complete and in-depth assessments than available for the current study.            

  1. Last question: Do children on drug therapy (particularly those on VPA therapy) have different transcript levels compared to other children in the autistic group?

We changed the figures. The use of colors allows the single values of the four patients under therapy to be visualized. We also added in the Discussion: “Valproic acid upregulates transcriptions of some HERV elements [109] and  antipsycotic drugs may induce epigenetic alterations [110]. No particularly aberrant expressions of HERVs or of TRIM28/SETDB1 were noted in the two children on valproic acid or in the two on other medications whose impact on these variables is unknown.”  

Reviewer 2 Report

The Paper from Tovo et al with the title “Enhanced expression of human endogenous retroviruses, TRIM28 and SETDB1 in autism spectrum disorder” 
assessed transcript levels of several HERV families as well as TRIM28 and SETDB1 in blood samples of children with autism spectrum disorders (ASD) beside healthy controls. They identified an increase in transcript levels for most HERV elements beside TRIM28 and SETDB1 in ASD patient samples.

Unfortunately, the manuscript requires additional key experimental results before publication.

Major Comments:

  1. Pol Levels of HERV-H, HERV-K and HERV-W groups were investigated. However, as the authors hypothesize that the inflammation triggered by HERV-derived proteins might impact ASD it is also essential to look at env levels of the described HERV groups. Especially HERV-K env as well as HERV-W env have been described in multiple recent studies to contribute to inflammation as well as impact neurological diseases (PMID: 29551251, PMID: 35063077). These need to be included.
  2. Beside Inflammatory markers should be investigated in the samples. It will be interesting to see if there is a positive correlation between inflammatory markers and HERV expression, which would strengthen the assumption that HERVs might paly a role on inflammatory processes in ASD patients.
  3. Some children were treated with medications like valproic acid. Valproic acid is known to induce some HERV groups including HERV-W env elements (PMID: 22253875). Thus it should be investigated if these two patients behave different in their HERV expression levels. Moreover, it should be stated if anything is known about the other drugs in respect to HERV activation.

Minor Comments:

  1. TRIM28 transcript levels are upregulated in ASD patients and are positively correlated with HERV levels. As the authors already describe/discuss TRIM28 is a key regulator of HERV expression. Thus high TRIM28 levels would/should result in low HERV levels as also mentioned by the authors. However, it has been shown that sumoylation of TRIM28 (PMID: 31391303) is essential for HERV regulation. Thus it would be nice to include Western Blots if samples are available to show maybe differences on TRIM28 sumoylation, which could explain the opposite data observed. If samples are not available this scenario should at least be discussed in the discussion section.
  2. All mentioned “data not shown” should be included as supplementary figures as they would strengthen the manuscript.
  3. Is HERV-W env meant with MSRV. If so, this should be explained in more detail at some point, otherwise it could appear that several HERV groups are being referred to here.
  4. Section 2.1. The abbreviation HC should be explained when used the first time.

Author Response

Reviewer 2

The Paper from Tovo et al with the title “Enhanced expression of human endogenous retroviruses, TRIM28 and SETDB1 in autism spectrum disorder” assessed transcript levels of several HERV families as well as TRIM28 and SETDB1 in blood samples of children with autism spectrum disorders (ASD) beside healthy controls. They identified an increase in transcript levels for most HERV elements beside TRIM28 and SETDB1 in ASD patient samples. Unfortunately, the manuscript requires additional key experimental results before publication.

Major comments

1.Pol Levels of HERV-H, HERV-K and HERV-W groups were investigated. However, as the authors hypothesize that the inflammation triggered by HERV-derived proteins might impact ASD it is also essential to look at env levels of the described HERV groups. Especially HERV-K env as well as HERV-W env have been described in multiple recent studies to contribute to inflammation as well as impact neurological diseases (PMID: 29551251, PMID: 35063077). These need to be included.

Studies on transcription levels of HERV-H-env, HERV-K-env, and HERV-W-env families in ASD have been published (Ref. 15,56). We assessed the pol genes of HERV-H, -K, and -W families 1) because they had not been investigated in ASD and thus they provide a new information; 2) to verify if the phenomenon of HERV activation in ASD, as underlined in the Conclusion, “…does not derive from the upregulation of a single HERV determinant, but from a more extensive and complex process” involving several HERV sequences. Please note that we assessed two specific envelope genes of the HERV-W family (Syncytin 1 and Multiple Sclerosis Associated Retrovirus à discovered and widely studied by the group of your PMID: 29551251 corresponding to the Ref. 38), and one envelope gene of HERV-FDR family (Syncytin 2) [see also point 6].  The impact of these env proteins on inflammation, immune response, and neurological disorders are more extensively and precisely documented (many targeted references are reported in the article) than the generic HERV-H-env, -K-env, and -W-env  families, which encompass a large number of distinct HERV sequences (e.g. about 100 for HERV-K-env).  

  1. Beside Inflammatory markers should be investigated in the samples. It will be interesting to see if there is a positive correlation between inflammatory markers and HERV expression, which would strengthen the assumption that HERVs might play a role on inflammatory processes in ASD patients.

On the one side, we underlined that “Activation of NF-kB and production of pro-inflammatory cytokines can induce HERV transactivation [58] and high concentrations of inflammatory cytokines were observed in blood and cerebrospinal fluid of autistic patients [14].” Furthermore ”…abnormal cytokine profiles were found to be associated with upregulation of HERVs in PBMCs from autistic children and their mothers [15].”  On the other side,HERVs can, in turn, shape the immune system and induce inflammatory reactions [16,27,34], with activation of the inflammasome [24]. The result may be a vicious circle leading to inflammatory-driven deleterious effects on the developing brain with consequent clinical manifestations of ASD [10-13].” Therefore, a positive correlation between inflammatory markers and HERV expressions could further strengthen the current data, but it  would not solve the dilemma: Which one comes first? Based on this, it was not among the goals of our study.

  1. Some children were treated with medications like valproic acid. Valproic acid is known to induce some HERV groups including HERV-W env elements (PMID: 22253875). Thus it should be investigated if these two patients behave different in their HERV expression levels. Moreover, it should be stated if anything is known about the other drugs in respect to HERV activation.

We have followed this useful suggestion. We changed the figures. The use of colors allows the single values of the four patients under therapy to be visualized. We also added in the Discussion: “Valproic acid upregulates transcriptions of some HERV elements [109] and  antipsycotic drugs may induce epigenetic alterations [110]. No particularly aberrant expressions of HERVs or of TRIM28/SETDB1 were noted in the two patients on valproic acid or in the two on medications whose impact on these variables is unknown.”  

Minor comments

  1. TRIM28 transcript levels are upregulated in ASD patients and are positively correlated with HERV levels. As the authors already describe/discuss TRIM28 is a key regulator of HERV expression. Thus high TRIM28 levels would/should result in low HERV levels as also mentioned by the authors. However, it has been shown that sumoylation of TRIM28 (PMID: 31391303) is essential for HERV regulation. Thus it would be nice to include Western Blots if samples are available to show maybe differences on TRIM28 sumoylation, which could explain the opposite data observed. If samples are not available this scenario should at least be discussed in the discussion section.

Unfortunately, samples are not available for additional tests using Western Blots. As regards the importance of sumoylation, in the Discussion we underlined: “TRIM28 is a small ubiquitin-related modifier (SUMO) that, through binding to lysine residues of target proteins, causes their phosphorylation and proteasome-driven degradation. TRIM28 recruits SETDB1 for SUMOylation, a crucial transient post-translational event involved in essential cell functions, such as transcriptional repression, RNA splicing, and protein degradation [92,93]. Subsequently, now we add: “It must be remembered that the SUMOylation induced by TRIM28/SETDB1 and their potential functional interactions with HERVs may be regulated by post-translational events between the encoded proteins, whereas we assessed only their transcriptional profiles.”

  1. All mentioned “data not shown” should be included as supplementary figures as they would strengthen the manuscript.

We have followed this suggestion and added specific supplementary figures instead of data not shown.

  1. Is HERV-W env meant with MSRV. If so, this should be explained in more detail at some point, otherwise it could appear that several HERV groups are being referred to here.

In the Introduction we have clarified“….some viral sequences are transcribed and a few encode proteins, such as Syncytin 1 (SYN1), an  envelope protein encoded by HERV-W [17], and Syncytin 2 (SYN2), an envelope protein encoded by HERV-FDR [18], which have been coopted for essential physiological functions. For instance, they are engaged in placental syncytiotrophoblast formation and regulation of crucial immune functions [19-22].” We also add: “Furthermore, some viral proteins, such as the envelope protein encoded by HERV-W and referred to as multiple sclerosis-associated retrovirus (MSRV), can trigger brain inflammation and autoimmunity [28-30],….”

  1. Section 2.1. The abbreviation HC should be explained when used the first time.

OK, thanks. We explained the abbreviation HC when used the first time in Section 2.1.

Reviewer 3 Report

The authors of the article present a very straightforward approach showing that HERV related genes are differentially expressed between ASD and HC. 

The overall presentation is very sound and clear and the findings are of great interest. I thus only have minor suggestions for improvements. 

In the introduction, the authors state that environmental factors play an important role. In ASD however, twin studies suggest otherwise. I would suggest explicitly mentioning how high the effect of the shared/non-shared environment has been estimated so far. 

Also, it is a bit unclear which environmental factors would trigger HERV activation and if/how they are related to ASD. The authors actually discuss that it is unclear if the observation presented is causally related to ASD or a consequence. 

Show a descriptive table of the cohorts, showing the age distribution, medication, ADOS scores etc. 

The results are presented very shortly. I would suggest adding the p-values and effect sizes in the text. The figures should be improved in that the median is visible. It disappears among the squares (maybe use grey boxes) 

some parentheses of the references are not squared parentheses but round ones. 

Author Response

Reviewer 3

The authors of the article present a very straightforward approach showing that HERV related genes are differentially expressed between ASD and HC. The overall presentation is very sound and clear and the findings are of great interest. I thus only have minor suggestions for improvements.

We appreciate very much the referee’s interest in our study and her/his positive comment.

1.In the introduction, the authors state that environmental factors play an important role. In ASD however, twin studies suggest otherwise. I would suggest explicitly mentioning how high the effect of the shared/non-shared environment has been estimated so far.

In the introduction, after the phrase on the results emerged from twin studies, we now point out: “Increasing evidence underlines, however, the importance of environmental factors, particularly during the critical phase of central nervous system development [4,5].” Then, please note that the following six lines of the introduction and 9 references focus on environmental factors and biological mechanisms potentially involved. In addition, in the Discussion there is a large space on single environmental factors and their possible pathogenetic mechanisms (see point 2). Considering that a precise quantification of the impact  of environmental factors on the development of ASD in genetically predisposed individuals remains poorly defined in literature, we fear that a further in-depth analysis of this topic might disperse attention in the context of our work.   

  1. Also, it is a bit unclear which environmental factors would trigger HERV activation and if/how they are related to ASD. The authors actually discuss that it is unclear if the observation presented is causally related to ASD or a consequence.

Among environmental factors that could trigger HERV activation and their possible relation with ASD in the Discussion we underline:Exposure of pregnant women to infectious agents induces activation of the maternal immune system with increased risk for neuropsychiatric disorders in their children, including ASD [12,59-61]. The stimulation of the immune system may result in increased production of neuroinflammatory cytokines in the offspring, with consequent insult to the central nervous system, alterations in synaptic protein expression, and abnormal synaptic connectivity, all of which are typically found in ASD [9,62]. Some  exogenous viral infections can trigger HERV activation [63-69], which, for example in the case of CMV, may also give rise to asymptomatic forms during pregnancy.” Subsequently we add: “Prenatal and early postnatal exposure to  negative environmental factors, such as pollutants, chemical agents, infections, immune activation, and epigenetics have been associated with ASD [111-113]. These environmental factors, e.g. exposure to pesticides [114], maternal cigarette smoking [115], infective agents [63-69], and immune stimulation [58], exhibit a significant impact on HERV expression.”

As regards the possible causative role of HERV activation in ASD, the association between HERV overexpression and inflammatory/autoimmune diseases  and neurologic disorders is widely documented. Experimental and in animal studies support the etiopathogenetic role of HERVs in these disorders. Nevertheless, whether HERVs  are causative or secondary effects  of these diseases remains an unsolved dilemma, and we underlined this general doubt in the discussion.

  1. Show a descriptive table of the cohorts, showing the age distribution, medication, ADOS scores etc.

We have added Table 1 to summarize demographics and clinical characteristics of the autism sample.

  1. The results are presented very shortly. I would suggest adding the p-values and effect sizes in the text.

                We followed this suggestion and now p-values and effect sizes are reported in the text.

  1. The figures should be improved in that the median is visible. It disappears among the squares (maybe use grey boxes).

We thank you very much for this useful comment. We changed the figures and now, using  grey boxes, the median is visible. In addition, the use of colors allowed to identify  the single values of the four patients on therapy.

  1. Some parentheses of the references are not squared parentheses but round ones.

OK, Thanks. We corrected the round parentheses.